# Effect of Particle Strength on SiCp/Al Composite Properties with Network Architecture Design

**DOI:** 10.3390/ma17030597

**Published:** 2024-01-26

**Authors:** Xiang Gao, Xiaonan Lu, Xuexi Zhang, Mingfang Qian, Aibin Li, Lin Geng, Huan Wang, Cheng Liu, Wenting Ouyang, Hua-Xin Peng

**Affiliations:** 1Ningbo Innovation Center, Zhejiang University, Ningbo 315100, China; gaoxiang1986@zju.edu.cn (X.G.); xnlu@zju.edu.cn (X.L.); 2Institute for Composites Science Innovation (InCSI), School of Materials Science and Engineering, Zhejiang University, Hangzhou 300027, China; hwang2014@zju.edu.cn (H.W.); 11926075@zju.edu.cn (C.L.); 12126008@zju.edu.cn (W.O.); 3School of Materials Science and Engineering, Harbin Institute of Technology, Harbin 150001, China; mingfang.qian@hit.edu.cn (M.Q.); aibinli@hit.edu.cn (A.L.); genglin@hit.edu.cn (L.G.)

**Keywords:** metal matrix composites (MMCs), finite element analysis (FEA), interface, micro-mechanics, network architecture

## Abstract

Recent works have experimentally proven that metal matrix composites (MMCs) with network architecture present improved strength–ductility match. It is envisaged that the performance of architecturally designed composites is particularly sensitive to reinforcement strength. Here, reinforcing particles with various fracture strengths were introduced in numerical models of composites with network particle distribution. The results revealed that a low particle strength (1 GPa) led to early-stage failure and brittle fracture. Nevertheless, a high particle strength (5 GPa) delayed the failure behavior and led to ductile fracture at the SiC/Al–Al macro-interface areas. Therefore, the ultimate tensile strengths (UTS) of the network SiC/Al composites increased from 290 to 385 MPa, with rising particle strength from 1 to 5 GPa. Based on the composite property, different particle fracture threshold strengths existed for homogeneous (~2.7 GPa) and network (~3.7 GPa) composites. The higher threshold strength in network composites was related to the increased stress concentration induced by network architecture. Unfortunately, the real fracture strength of the commercial SiC particle is 1–2 GPa, implying that it is possible to select a high-strength particle necessary for efficient network architecture design.

## 1. Introduction

Discontinuously reinforced metal matrix composites (MMCs) often possess a homogeneous distribution of reinforcements [1]. Unfortunately, it always sacrifices elongation to enhance the strength [2]. The inversion relationship between strength and toughness is a bottleneck problem of MMCs. Actually, it has been proposed that inhomogeneous reinforcement distribution might induce enhanced strength, as well as ductility [3,4]. The co-continuous [4,5], interpenetrating [6], and network distribution [7,8,9] showed potential for coinstantaneous strengthening and toughening with isotropy. In recent years, simultaneously improved strength and toughness have been demonstrated in titanium [8] and aluminum [9] matrix composites with network whisker/particle distribution. Hence, inhomogeneous network architecture design is a novel and significant issue for MMCs.

For commercial SiC particles, whiskers or fibers, the fracture strength strongly depends on the preparation and heat treatment techniques [10]. The X-ray CT data and finite element method (FEM) indicate that the reinforcement-rich regions are potential sites for crack initiation [11]. In our previous numerical work, the equivalent plastic strain contour implied that particle/whisker fractures in the network layer may be the main cause for the early-stage failure of the SiC/Al network composite [12]. Thus, the strength of reinforcement may be an important factor for the design of inhomogeneous MMC architecture.

It is usually accepted that, for homogeneous composites, the high-strength SiC particle is the premise of high performance. However, micro-cracks are primarily initiated by the matrix failure in homogeneous SiCp/Al composites [13,14], implying that even stronger reinforcement provides limited enhancement. On the other hand, Segurado et al. indicated that the sensible fracture strength of the SiC particle is in the range of 1–2 GPa [15], which is widely applied in SiCp/Al simulation works [12,16,17,18]. However, Zhang et al. ignored SiCp brittle fracture in their FEM work, based on the experimental results that showed that few particles fractured [14]. Considering that an inhomogeneous reinforcement distribution may induce stress concentration in particle-rich areas [12], structural design may desire higher-strength reinforcement. Therefore, it is necessary to study the difference between the effects of particle strength on the deformation and fracture behaviors in homogeneous and network composites.

However, applying SiC particles with various fracture strengths in the experience requires much more time consumption. The finite element method simulation is an adaptive choice. It has been a mature method used to predict deformation and failure behaviors of homogeneous MMCs and is widely applied to investigate the effects of clustering [18], aspect ratio [19], orientation [20], particle shape [21], etc. In addition, the FEM was developed to predict the mechanical behavior of composites with architecture design, i.e., woven fiber [22], interpenetrating structure [23], honeycomb architecture [24], biomimetic scaffold [25], and masonry structure [26]. Unfortunately, the traditional numerical models of architected composites were too small [27] or over-simplified [28]. Hence, the results presented 20% anisotropy [27] and a lack of representativeness [28]. It is evident that an adaptable model with network architecture design is required.

Recently, we developed a novel digital modeling technology to establish a representative volume (RVE) model of the discontinuous network distribution of particles/whiskers [12,16,17]. It was successfully applied to investigate the effects of particle shape [17], aspect ratio [12], and particle size ratio [16] in the network composites. This method was accepted and followed by other researchers [29]. Therefore, the novel simulation method is appropriate for the investigation of the effects of particle strength on the deformation and fracture behaviors in architected composites.

In the present work, we established a novel composite model with network-distributed SiC particles and a homogeneous counterpart. The various particle strengths were introduced in the models. The stress–strain curves were predicted via FEM simulation. The deformation and fracture behaviors were studied by analyzing the stress/strain field contours.

## 2. Finite Element Method (FEM)

### 2.1. Micro-Structural Modeling

The previous experimental observation presented that a real network architecture is comparable to grain boundaries in metals [7]. Thus, a developed Voronoi arithmetic was applied to form ‘grain boundary’-like network architecture [30,31]. Too many network cells and periodic geometry greatly increase the complexity of the models. Thus, more and more element mesh is required, which is hard to be solved by the FEM code. Therefore, an 8-cell network architecture without periodicity was established. Then, equiaxial polyhedron SiC particles were arranged on the network layers (Figure 1a–d). The model edge length was 100 μm, which is approximately 10 times the SiC particle diameter. The overall volume fraction of the SiC particle was 10%, while the particle-rich region had a local volume fraction as high as 56%. The geometry model was meshed by the 4-nodes tetrahedron element (C3D4) (Figure 1e). Approximately 820,000 elements were meshed in the model. Due to the lack of periodicity in the geometry, the uniaxial tensile boundary condition was applied. These boundary conditions were defined as follows: The surfaces x = 0 and x = 100 kept parallel to the plane YOZ. Other surfaces were constrained by a similar operation. The displacement load was applied on the ‘RF’ with a maximum tensile strain of 8% along the *x*-axis (see Figure 1f). The statistics result showed that the network plane-included angle was concentrated at 120° (Figure 1g). The simulation was solved by the ABAQUS explicit module.

### 2.2. Material Properties

In the models, the matrix was 6061Al-T6, with a density of 2.70 g/cm^3^, Young’s modulus of 68.9 GPa, Poisson’s ratio of 0.33, and a yield strength of 269.1 MPa (Appendix A). The failure behaviors were introduced in the models. The tetrahedron element experiencing failure implied the complete loss of load-carrying capacity. Thus, a progressive degradation of the material stiffness model was applied in the simulation on failure behaviors [32,33] (Appendix A):(1)$$E=\left(1-D\right)\bar{E}$$
where *D* is the stiffness degradation variable, *E* is the Young’s modulus with degradation to represent the stiffness, and $\bar{E}$ is the undamaged stiffness tensor. Therefore, the stress tensor showed the same form:(2)$$\sigma =\left(1-D\right)\bar{\sigma }$$
where *σ* is the stress tensor with degradation, and $\bar{\sigma }$ is the undamaged stress tensor. The ductile failure behavior of the matrix was expressed by the ‘ductile damage criterion’, in which cracks were initiated when the equivalent plastic strain *ε^p^_f_* reached 0.11 (determined by the experimental stress–strain curve in Appendix A). In addition, the total plastic strain increment after crack initiation (*ε^d^*) was assumed to be 0.001. The linear stiffness degradation behavior was used in the matrix, which is given by [33]:(3)$$D_{Al}={\dot{\varepsilon }}^{p}/\varepsilon^{d}$$
where ${\dot{\varepsilon }}^{p}$ is the equivalent plastic strain of the element after damage initiation. The SiC particle had a density of 3.2 g/cm^3^, Young’s modulus of 427.0 GPa, and Poisson’s ratio of 0.17. We introduced ‘brittle failure criterion’ in SiC particles. The crack initiates when the maximum principal tensile stress exceeds the fracture strength of the SiC particle *σ^p^_f_* (Mode I cracking). The shear behavior (Mode II cracking) is considered after crack initiation, in which shear retention can be defined as a power law [34]:(4)$$\rho \left(e^{ck}\right)={\left(1-\frac{e^{ck}}{e_{max}^{ck}}\right)}^{p}$$
where *ρ* is the shear retention factor depending on the crack opening strain *e^ck^*, and *p* and *e^ck^_max_* are material parameters. Therefore, the stiffness degradation behavior of ceramic reinforcement can be expressed by [34]:(5)$$D_{SiC}=\frac{\rho }{1-\rho }G\;-$$
where *G* is the shear modulus of the SiC particle. Zhang et al. numerically extrapolated the parameters by fitting the stress–strain curve in the 7% SiCp/Al composite and found that *e^ck^_max_* = 0.2, and *p* = 2 [35]. Different fracture strengths of the SiC particle *σ^p^_f_* were assumed in the model, including 1, 2, and 5 GPa. As previous experimental fracture surfaces showed a retained matrix on the particle surface (Appendix A) in network [36] and homogeneous [13] SiCp/Al composites and thus confirmed the strong interfacial bonding state, a perfect interfacial cohesion was introduced in present models.

## 3. Result and Discussion

The finite element (FE) model was validated in our previous work [12]. The predicted stress–strain curves and mechanical properties of 10, 15, and 20 vol.% SiCp/6061Al composites were close to the experimental counterparts [37,38,39] (Appendix A). In addition, the stability of the models was verified by comparing the simulation results of three 10 vol.% SiCp/6061Al geometries [12] (Appendix A). Therefore, our numerical method is reliable for predicting deformation and fracture behaviors of SiC/Al composites.

The main crack propagation of the network composite was presented by contours of equivalent plastic strain (see Figure 2). In the composite reinforced with 1 GPa SiC particles (Figure 2a), early-stage cracks initiated even at the elastic deformation stage when the particle-lean regions were just subjected to elastic strain (marked A in Figure 2a, plastic strain is 0). Arrow B shows a typical brittle fracture characteristic in the fracture surface, where the plastic strain was only ~0.055, much lower than that for crack initiation in the matrix, where *ε^p^_f_* = 0.11.

In the composite with 2 GPa particles, brittle fracture also existed (C in Figure 2b). The early-stage crack initiation was attributed to very small inter-particle distance, since the local content of the SiC particle was very high (~56%). However, ductile fracture occurred in junction layers (D in Figure 2b). The crack propagated along the SiC/Al–Al macro-interface areas. Cracks were also observed in the layers parallel to the loading direction (rectangle and circle in Figure 2b).

On the other hand, in the composite with the 5 GPa particle, the main crack was produced in the matrix, since the SiC particle did not fracture (E in Figure 2c). The produced crack propagated along the SiC/Al–Al macro-interface and reached parallel layers (F in Figure 2c). Obviously, such crack initiation and propagation behaviors are favorable for crack deflection and thus help in toughening the composite.

**Figure 2 materials-17-00597-f002:**
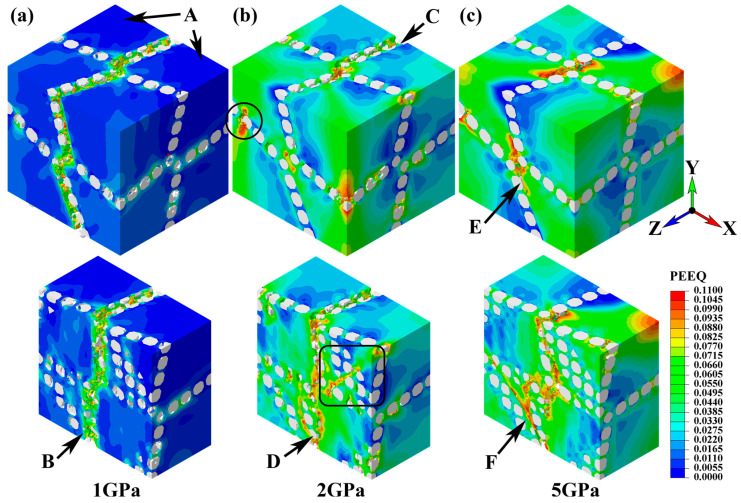
The equivalent plastic strain contours showing crack initiation and propagation behavior in the SiC/Al network composites at a extern strain (*ε_xx_*) of 4.0% with various particle strengths: (**a**) 1 GPa; (**b**) 2 GPa; and (**c**) 5 GPa. Arrow A: pure elastic deformation (plastic strain is 0). B and C indicate brittle fracture along perpendicular layers showing a low-strain fracture surface (~0.055). D: ductile fracture in junction layers with sufficient plastic strain (~0.110). E and F show ductile fracture along the SiC/Al–Al macro-interface.

From the viewport of the micro-crack, to toughen the architected composite, the large monolithic reinforcement-lean area and thin reinforcement-rich layer are required. During application of the tensile stress, the mode I crack dominated the failure behavior of the network composite. The stress at the crack tip (*σ_tip_*) was directly proportional to crack length (*a*) and inversely proportional to the distance to it (*r*):(6)σtip∝ar

Due to stress concentration in the reinforcement-rich layer, the micro-crack initiates in the layers parallel to the external load direction. The length of the mode I micro-crack depends on the thickness of the layer. A large monolithic reinforcement-lean area implies an increasing *r*. Thus, the primary micro-crack in the network layer tends to be blunted by neighboring matrix cells (circle in Figure 2b). From the viewport of the macro-crack, the crack branching (rectangle in Figure 2b) and deflection (F in Figure 2c) lead to a zigzag crack morphology and thus a longer propagation distance. These behaviors cost more energy. In addition, zigzag propagation means the crack is not perpendicular to the external load. Our previous theoretical analysis showed that the crack propagation resistance increases dramatically with decreasing orientation between the crack and the external load [36]. This analysis is fixed with the experimental observations that the network architecture design promotes primary micro-crack blunting [40,41], macro-crack deflection [42,43], and branching [42,43]. Therefore, the crack propagation is delayed with more energy consumption. This is beneficial to the toughness of the composites.

Our previous work [12] demonstrated that ductile fracture occurred in network SiCw/Al and homogeneous SiCp/Al composite models. The ductile fracture behavior led to the high plastic strain fracture surface and neighboring low strain gradient. When the particle strength increased from 1 to 5 GPa, the plastic strain on the fracture surface increased from ~0.055 to ~0.110, and the strain gradient decreased from ~0.022 μm^−1^ to ~0.006 μm^−1^ (areas marked B, D, and E). Therefore, the fracture behavior of the network composite changed from brittle at the network layer (SiC strength 1 GPa) to ductile fracture along the SiC/Al–Al macro-interface (SiC 5 GPa). However, once the failure mechanism is dominated by the ductile fracture along the SiC/Al–Al interface, further increasing SiC strength cannot enhance the composite strength because sufficient strength is achieved to resist fracture.

Comparing the stress–strain curves of composites with network and homogeneous distributions of particles (i.e., plots 2 and 4 in Figure 3), network architecture provided an obvious increment on elastic modulus and yield strength. The network composite with a 2 GPa fracture strength of the SiC particle exhibited higher modulus and strength: the modulus increased from 86 GPa to 90 GPa, and the yield stress changed from 299 MPa to 315 MPa. This is related to the fact that the clustering particle in network architecture led to a high local content of reinforcement and thus increased stress concentration. As a result, more load was transferred from the matrix to the particles, and the load-bearing efficiency of network-arranged reinforcement was improved. However, the higher stress at particle-rich areas implies earlier fracture. Hence, the ultimate tensile strength (UTS) of the network composite was limitedly enhanced. Moreover, the increased UTS in the network composite often sacrifices the elongation and toughness, which is consistent with the effect of clustering particles [11].

However, the network composite with 5 GPa SiC showed different characteristics. The UTS of the 5 GPa composite (385 MPa) was much higher than that of the 1 GPa (290 MPa) and 2 GPa (330 MPa) counterparts. More interestingly, the composite with the 5 GPa particle presented a much larger strain at UTS point, i.e., *ε_xx_* = 2.4% (marked D in Figure 3), compared to *ε_xx_* = 1.2% (2 GPa SiC, marked A) and *ε_xx_* = 0.38% (1 GP SiC, marked C). In addition, the fast stress drop stage—corresponding to the main crack initiation and propagation—had different slopes in these composites. This shows that the fracture mode of network composites does change with the particle fracture strength. The fracture of the 1 GPa composite was dominated by brittle fracture along the perpendicular SiCp layers (see Figure 4a), while the fracture was ductile in the 5 GPa composite along the macro-interface of SiC/Al–Al. The 2 GPa composite exhibited mixed brittle and ductile fracture characteristics. Therefore, after the initiation of the main crack (marked B, D, and E in Figure 3), the stress–strain curves of composites with 1 and 2 GPa particle strengths declined more dramatically than that of the 5 GPa composite. Also, reduced elongation still existed in the network composite with the 5 GPa particle. A possible strategy may be the replacement of the particle with a whisker [12], since the latter has a higher crack-resistant capacity than the former via crack bridging and deflection. Therefore, it is still an open question to pursue an ideal strengthening effect without sacrificing elongation (reduce toughness) in MMCs with network architectures.

The UTS of the network composites is determined by the load-bearing capacity of the SiC particles, since the ceramic particle presents much higher strength and modulus than the matrix alloy. The network architecture design presents increased modulus and strength [36], yet earlier fracture of the particle is predicted. As discussed, when particle strength is high (i.e., 5 GPa in this work), increasing the particle strength cannot further enhance the composite strength any more. In addition, higher SiC strength in network composites (see Figure 4) implies a higher threshold strength of the particle. That is to say, high reinforcement fracture strength is more significant for the network composite, implying that optimal particle strength is very important for the efficient design of network composites.

A feasible solution to evaluate the threshold particle strength was realized via simulating the composite properties with various particle strengths. According to the simulation result (see Figure 5), the threshold strength of the particle in the homogeneous composite was ~2.75 GPa, and the network composite was ~3.75 GPa. This gap is attributed to the higher stress level (i.e., load-bearing capability) in the SiC-rich region in network composites, due to the stress concentration led by high local content. Therefore, the dependence of the threshold strength of reinforcements on architecture can be quantitatively analyzed.

To quantitatively analyze the load-bearing capacity of reinforcement, the stress concentration factor *R* was applied [44]. From the perspective of material mechanics, the effective stress (*σ_eff_*) in the composite can be represented by the rule of mixture (ROM):(7)$$\sigma_{eff}=\left(V_{SiC}\sigma_{SiC}+V_{Al}\sigma_{Al}\right)/V_{composite}$$
where *V_SiC_* and *V_Al_* are the volumes of the reinforcement and matrix, respectively, *σ_SiC_* and *σ_Al_* are the average stresses, and *V_Composite_* is the total volume of the composite:(8)$$V_{composite}=V_{SiC}+V_{Al}=\sum_{i=SiC}V_{i}+\sum_{i=Al}V_{i}$$
where *V_i_* is the stress in each element of the reinforcement or matrix. The average stress is given by:(9)$$\sigma_{SiC}=\frac{\sum_{i=SiC}V_{i}\sigma_{i}}{V_{SiC}}$$
(10)$$\sigma_{Al}=\frac{\sum_{i=Al}V_{i}\sigma_{i}}{V_{Al}}$$

The stress concentration factor (*R_SiC_*, *R_Al_*) is applied to describe the inhomogeneity distribution of stress in the reinforcement and matrix, which is defined as [45]:(11)$$R_{SiC}=\frac{\sigma_{SiC}}{\sigma_{eff}}$$
(12)$$R_{Al}=\frac{\sigma_{Al}}{\sigma_{eff}}$$

By introducing the volume fraction of SiC (*f_SiC_*) and Al (*f_Al_*), Equation (7) can be modified to:(13)$$\sigma_{eff}=f_{SiC}\sigma_{SiC}+f_{Al}\sigma_{Al}$$

Combining Equations (11)–(13), the relationship between *f* and *R* is expressed as:(14)$$1=f_{SiC}R_{SiC}+f_{Al}R_{Al}$$

Therefore, for a fixed composite system, the relationship between *R_SiC_* and *R_Al_* is constant. That is to say, investigation of *R_SiC_* is sufficient.

The *R_SiC_* evolution curves are presented in Figure 6. Comparing the curves of Figure 6a, the *R_SiC_* and its increasing slope were higher in the network composite during the plastic deformation. Therefore, it can be deduced that the network architecture design promotes the load-bearing capacity of the reinforcement. It is because the clustered particle bears more stress than the homogeneous counterpart [46]. However, a higher load-bearing capacity indicates the early particle fracture of SiC particles. Thus, the *R_SiC_* of the network architecture design dropped earlier.

Comparing the curves of Figure 6b, the *R_SiC_* of the network composite with the 5 GPa particle showed an obviously different evolution. The particle was sufficiently strong to resist fractures. Therefore, a long plastic stage was presented. After *ε_xx_* = 2.6%, the void coalescence occurred in the matrix. Due to no particle fracture, the load-bearing capacity of reinforcement remained stable. Thus, the *R_SiC_* curve stopped increasing and remained constant. By contrast, the 2 GPa curve presented a clear decreasing evolution, owing to particle fracture, resulting in a decreasing load-bearing capacity.

**Figure 6 materials-17-00597-f006:**
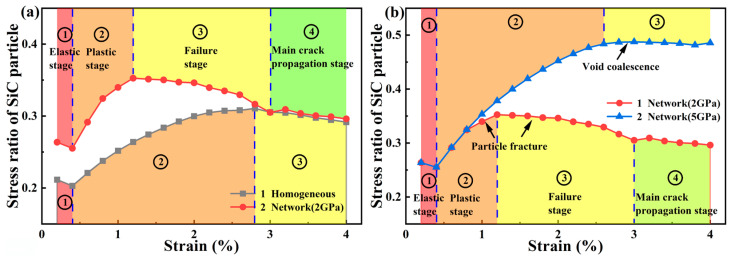
The *R_SiC_* evolution curves of SiCp/6061Al composites with homogeneous and network architectures. (**a**) The comparison between homogeneous and network composites with the 2 GP SiC particle. (**b**) The comparison between network composites with 2 GP and 5 GPa SiC particles.

In our previous work, the effect of particle size ratio (PSR) was investigated in the network SiCp/Al composite [16]. High PSR presented an increased modulus, due to the enhanced load-bearing capability of SiC (see Appendix A) because the local volume fraction of SiCp increased with PSR and thus carried higher loads at SiC-rich regions (see Figure 4b,c). Therefore, the threshold strength of SiC was even higher at a higher PSR. In addition, our simulation results presented that the stress level of reinforcement in mono-layer composites increased with the orientation angle (see Figure 7). It agreed that the stress in parallel layers was much higher than that in perpendicular ones (see Figure 4b,c). Lee et al. [47] designed a network architecture but with a reduced modulus, as a large fraction of network layers were 45° or 90° to the external load (see Appendix A). In this case, the threshold strength of particles was low. Conversely, laminated [48], bar-like [49], and ring-like [50] (see Appendix A) architecture may show high threshold strength when particles aligned parallel to the external load.

In some previous SiCp/Al models, particle fracture was not considered, since it is usually assumed that particles are strong enough to resist fracture [14]. This was also verified by our previous experimental work on the SiCp/Al network composite, in which cracks were principally initiated by matrix failure or interfacial decohesion and not the particle fracture [36] (see Appendix A). Hence, after the particles reach the threshold strength, it is not necessary to pursue overly high strength.

For homogeneous composites, the fracture of particles takes place when an external stress exceeds the Griffith criterion [51], where the fracture stress is given by:(15)$$\sigma_{c}^{p}=K/\sqrt{d}$$
where *K* is the fracture toughness and the geometrical factors-related constant, and *d* is the average diameter of the particle. Flom and Arsenault [52] found that particle fracture dominated the crack initiation and propagation behaviors in the coarse particle reinforced SiCp/Al composite. Experimental work proved that network architecture design was unsatisfactory in the SiCp/Al composite with a particle diameter of 10 μm [36] but was favorable for nano Al_2_O_3_/Al with fine (~2 μm) particles, based on the strength–ductility match [41]. The fine ceramic particles exhibited a higher fracture strength, due to the lower defect density.

## 4. Conclusions

We established a 3D SiCp/6061Al model with novel network particle distribution. The effects of SiC particle fracture strength on failure behaviors and mechanical properties were numerically investigated. Based on the simulation results, the main conclusions can be drawn as follows:(1)The particle fracture strength was significant for the failure behavior of the SiCp/Al composite. Low particle fracture strength (1 GPa) led to network composite failure at the elastic stage (*ε_xx_* = 0.38%). Contours of equivalent plastic strain showed that the brittle fracture occurred along the network layer perpendicular to the load direction. In the composite with medium particle fracture strength (2 GPa), failure was initiated at the small plastic strain (*ε_xx_* = 1.2%), and network boundaries could slow the crack propagation rate. High particle fracture strength (5 GPa) led to late composite failure initiation (*ε_xx_* = 2.4%), which showed ductile fracturing along the ‘interface’ of SiC/Al–Al. When the failure mechanism became ductile fracture, further increasing particle fracture strength did not enhance the composite strength.(2)The particle fracture strength significantly affected the ultimate tensile strength (UTS) of network composites. The UTSs of composites reinforced by particles with low (1 GPa), medium (2 GPa), and high strengths (5 GPa) were 290, 330, and 385 MPa, respectively. The network composite with 1 GPa SiCp failed before the yield point, which was seriously harmful to the elongation. By contrast, the composites with 2 and 5 GPa SiCp failed at higher strains, resulting in more elongation of the composites.(3)Network particle distribution favored the composite strength. The stress concentration factor of SiCp indicated that the network particle distribution benefited the enhanced load-bearing capability of SiC particles, which improved the mechanical properties in the elastic/plastic stages. Compared to the homogeneous composite, the network architecture design with 2 GPa SiCp successfully enhanced the elastic modulus from 86 GPa to 90 GPa and the yield stress from 299 MPa to 315 MPa. However, the network design lost the strengthening effect when particle fracture strength was too low (1 GPa). It proved that a high enough fracture strength of reinforcement was the prerequisite for the efficient strengthening of the network design.(4)The threshold strength of the particle was different between homogeneous and network composites. The threshold value was ~2.75 GPa for the homogeneous SiCp/Al composite, while it increased to ~3.75 GPa for the network SiCp/Al composite because the network architecture promoted load-bearing capacity in reinforcement-rich regions. High local volume fraction and low aligning orientation of the particle increased the threshold value. Moreover, reduced particle size was favorable for a better strength–ductility match in the network composite.

## Figures and Tables

**Figure 1 materials-17-00597-f001:**
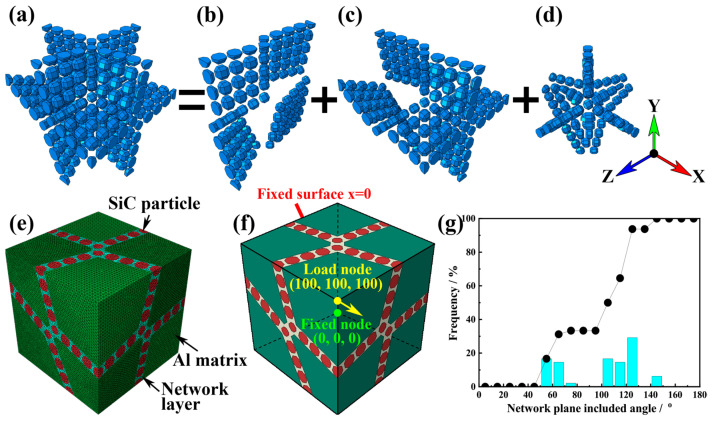
Geometrical model and mesh with network architecture. (**a**) SiCp arranged similarly to grain boundary structure; (**b**) SiCp layer perpendicular to the loading direction; (**c**) SiCp layer parallel to the loading direction; (**d**) junction layer linking perpendicular and parallel layers; (**e**) meshes for network composites; (**f**) boundary condition of the FEM model; (**g**) statistics on the angle between neighboring network layers in models with network architecture.

**Figure 3 materials-17-00597-f003:**
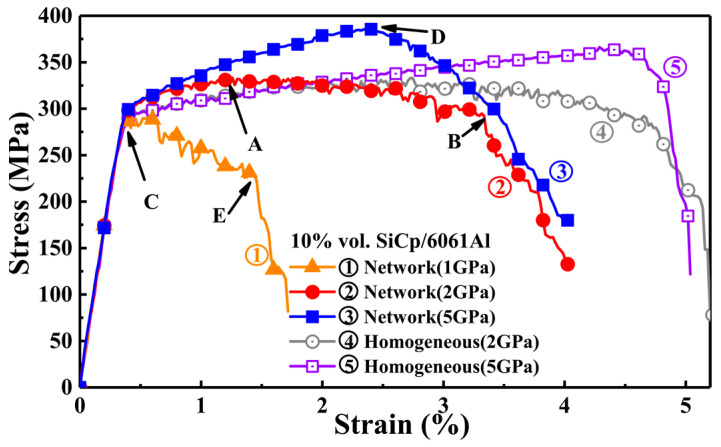
Tensile stress–strain curves of SiCp/6061Al composites with homogeneous and network architectures. The strengths of SiCp are 1, 2, and 5 GPa. A: network composite failure earlier than the homogeneous composite. B and E: main crack initiation. C: failure before yield strength; D: failure at *ε_xx_* = 2.4%.

**Figure 4 materials-17-00597-f004:**
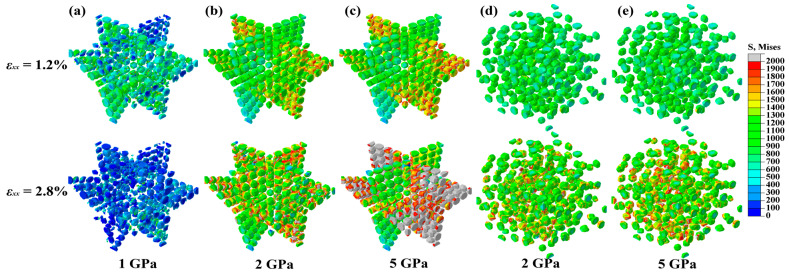
Von Mises stress contour of homogeneous (**a**–**c**) and network (**d**,**e**) composites with external strains reaching 1.2% and 2.8%, respectively. The particle fracture strengths are 1 GPa (**a**), 2 GPa (**b**,**d**), and 5 GPa (**c**,**e**), respectively.

**Figure 5 materials-17-00597-f005:**
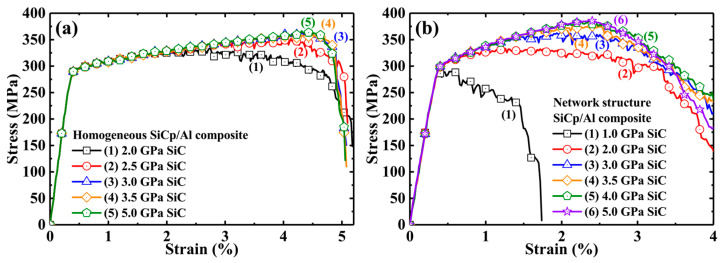
Stress–strain curves of (**a**) homogeneous and (**b**) network SiCp/Al composites with various reinforcement fracture strengths.

**Figure 7 materials-17-00597-f007:**
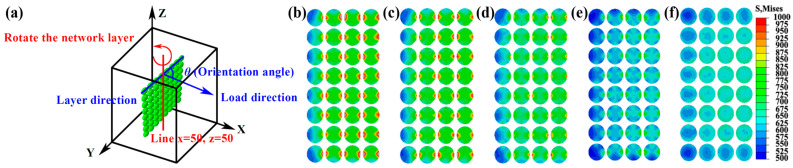
SiC particle-rich layer-reinforced Al matrix composite model (**a**) and the load-bearing state of SiC particles (**b**–**f**). The layer orientations are: (**b**) 0°; (**c**) 15°; (**c**) 20°; (**d**) 30°; (**e**) 45°; and (**f**) 90°, respectively.

## Data Availability

Some or all data, models, or code that support the findings of this study are available from the corresponding author upon reasonable request.

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
