# Peer review of "Effect of Particle Strength on SiCp/Al Composite Properties with Network Architecture Design"

_materials, 2024, doi:10.3390/ma17030597_

Round 1
Reviewer 1 Report
Comments and Suggestions for Authors
Dear Authors
Overall, I find the work to be valuable; however, I would like to share some constructive feedback and suggestions that I believe could enhance the clarity and impact of your research.
1. Elaboration in Introduction:
The introduction could benefit from a more elaborate presentation. Expanding on the study's background, significance, and potential implications would provide readers with a clearer context and better engage their interest.
2. Software Used for FE Modeling:
It would be beneficial to explicitly mention the software used for Finite Element (FE) modeling. Providing this information is crucial for readers to understand the methodology and potentially replicate or build upon your research.
3. Citations for Equations:
I noticed that some equations, such as Equation 1, lack proper citations. Ensuring that all equations are appropriately referenced will strengthen the credibility of your work and allow readers to trace the origins of key mathematical formulations.
4. Validation of FE Model:
One significant concern is the absence of experimental validation for the FE model. Without such validation, assessing the accuracy and reliability of the FE results becomes challenging. I recommend considering including experimental data or discussing potential limitations and uncertainties associated with the current model.
5. Improving Figure 2:
Regarding Figure 2, it has been noted that the contours representing PEEQ values are unclear.
I understand that these suggestions may require additional work, but I believe they will contribute to your article's overall quality and impact. I appreciate your effort in this research and look forward to seeing how these suggestions may be integrated into your work.
Comments on the Quality of English Language
The language looks fine.
Author Response
Reviewer 1's comments:
Comment #1: Overall, I find the work to be valuable; however, I would like to share some constructive feedback and suggestions that I believe could enhance the clarity and impact of your research.
Response: We thank the Reviewer for the constructive suggestions on our manuscript. We have revised the manuscript according to the comments and suggestions. The revisions were highlighted RED in the revised manuscript.
Comment #2: Elaboration in Introduction:
The introduction could benefit from a more elaborate presentation. Expanding on the study's background, significance, and potential implications would provide readers with a clearer context and better engage their interest.
Response: We thank the Reviewer for the suggestion on elaboration in Introduction Section. We expanded the contents of background, significance, and potential implications.
The introduction for inhomogeneous reinforcement distribution design is brief. So, we explain this issue in detail. The first paragraph “Discontinuously reinforced metal-matrix composites (MMCs) often possess homogeneous distribution of reinforcements[1]. Actually, it has been proposed that inhomogeneous reinforcement distribution might induce enhanced strength as well as ductility[2,3]. Simultaneously improved strength and toughness have been demonstrated in titanium[4] and aluminum[5] matrix composites with network whisker/particle distribution. Hence, inhomogeneous network architecture design is a novel and significant is-sue for MMCs” was modified to “Discontinuously reinforced metal-matrix composites (MMCs) often possess homogeneous distribution of reinforcements[1]. Unfortunately, it always sacrifices elongation to enhance the strength[2]. The inversion relationship between strength and tough-ness is a bottleneck problem of MMCs. Actually, it has been proposed that inhomogeneous reinforcement distribution might induce enhanced strength as well as ductility[3,4]. The co-continuous[5], interpenetrating[6] and network distribution[3,4,7] showed potential for coinstantaneous strengthening and toughening with isotropy. Recent years, simultaneously improved strength and toughness have been demonstrated in titanium[8] and aluminum[9] matrix composites with network whisker/particle distribution. Hence, inhomogeneous network architecture design is a novel and significant issue for MMCs” in the Manuscript at Line 29-38 Page 1.
Comment #3: Software Used for FE Modeling:
It would be beneficial to explicitly mention the software used for Finite Element (FE) modeling. Providing this information is crucial for readers to understand the methodology and potentially replicate or build upon your research.
Response: We thank the Reviewer for comment on the software. In this work we applied commercial software ABAQUS. The dynamic explicit FEM analysis was used to predict the deformation and fracture behaviors of the composites. We added the related description in the Manuscript at Line 98-99 Page 3: “The simulation was solved by the explicit module of commercial software ABAQUS”.
Comment #4: Citations for Equations:
I noticed that some equations, such as Equation 1, lack proper citations. Ensuring that all equations are appropriately referenced will strengthen the credibility of your work and allow readers to trace the origins of key mathematical formulations.
Response: We thank the Reviewer for the detailed comments on citations reference. We cited references of the citations on the description of Eq. 1-4: “Thus a progressive degradation of the material stiffness model was applied in the simulation on failure behaviors[34,35]:
|
|
(1) |
”, “The linear stiffness degradation behavior was used in the matrix, which is given by[35]:
|
|
(2) |
”, “The shear behavior (Mode II cracking) is considered after crack initiation, in which shear retention can be defined as a power law[36]:
|
|
(3) |
”, “Therefore, the stiffness degradation behavior of ceramic reinforcement can be ex-pressed by[36]:
|
|
(4) |
”
In addition, we added the references to the Reference List: “
- Alfarah, B., López-Almansa, F., Oller S. New methodology for calculating damage variables evolution in Plastic Damage Model for RC structures. Eng. Struct. 2017, 132, 70–86. https://doi.org/10.1016/j.engstruct.2016.11.022
- Li, F., Yuan, H., Liu, H. Implementation of metal ductile damage criteria in Abaqus FEA. J. Phys.: Conf. Ser. 2021, 1906, 012058. https://doi.org/10.1088/1742-6596/1906/1/012058
- Riahi, M.M. Numerical and experimental studies of the mechanical behaviour at the ice/aluminium interface. PhD Thesis, Université du Québec à Chicoutimi, Chicoutimi Canada, 2007.”
Comment #5: Validation of FE Model:
One significant concern is the absence of experimental validation for the FE model. Without such validation, assessing the accuracy and reliability of the FE results becomes challenging. I recommend considering including experimental data or discussing potential limitations and uncertainties associated with the current model.
Response: We thank the Reviewer for the comment on validation. We verified the reliability and stability of the FE models in our previous work. The numerical method was validated in our previous work[1] by comparing with predicted/experimental results of 10, 15 and 20vol.% SiCp/6061Al composites with homogeneous particle distribution[2-4] (see Fig. R1 and Table R1).
Fig. R1 Tensile stress-strain curves comparison between simulation[R1] and experiment[R2] results for the extruded 10vol.%, 20vol.% SiCp/6061Al composites and as-cast 15vol.% SiCp/6061Al composite with homogeneous particle distribution.
Table R1 Mechanical properties of the as-cast and extruded SiCp/6061Al composites by experiment and simulation[R1-4].
|
Materials |
Method |
E (GPa) |
σ0.2 (MPa) |
σUTS (MPa) |
εf |
|
10vol.% SiCp/6061Al (Extruded) |
Simulation[R1] |
86 |
345 |
382 |
5.4 |
|
Experiment[R2] |
84 |
342 |
375 |
5.3 |
|
|
15vol.% SiCp/6061Al (As-cast) |
Simulation[R1] |
96 |
318 |
353 |
- |
|
Experiment[R3] |
96 |
333 |
- |
- |
|
|
Experiment[R4] |
95 |
320 |
- |
- |
|
|
20vol.% SiCp/6061Al (Extruded) |
Simulation[R1] |
100 |
371 |
396 |
2.3 |
|
Experiment[R2] |
97 |
378 |
388 |
1.9 |
The stability of the models was confirmed by simulation on three 10vol.% SiCp/6061Al composites[1] (Fig. R2). Therefore, it is verified that our numerical method is reliable to predict deformation and fracture behavior of SiC/Al composites.
Fig. R2 The stress-strain curves of 10 vol.% SiCp/6061Al homogeneous composites with different SiCp distributions[R1].
The description of validation was added in the Manuscript at Line 138-143 Page 4: “The finite element (FE) model was validated in our previous work[17]. The predicted stress-strain curves and mechanical properties of 10, 15 and 20vol.% SiCp/6061Al composites are close to the experimental counterparts[13,39,40] (Figure S4, Table 1). In addition, the stability of the models was Verified by comparing simulation results of three 10vol.% SiCp/6061Al geometries[17] (Figure S5). Therefore, our numerical method is reliable to predict deformation and fracture behavior of SiC/Al composites”. The Fig. R1, R2 and Table R1 were added to the Supplementary Materials Page 3-4 as Fig. S4, S5 and Table S1.
Comment #6: Improving Figure 2:
Regarding Figure 2, it has been noted that the contours representing PEEQ values are unclear.
Response: We thank the Reviewer for the comment on improving figure. We agreed that the Figure 2 is not clear. This figure is so small. Thus we extract larger and clearer PEEQ contours in the ABAQUS. Then these contours were applied to combine into a new Figure 2 in the Manuscript at Line 182 Page 5, as shown in Fig. R3.
Fig. R3 The equivalent plastic strain the contour shows that crack initiation and propagation behavior in the SiC/Al network composites at strain (εxx) 4.0% with various particle strength: (a) 1 GPa; (b) 2 GPa; (c) 5 GPa. Arrow A: pure elastic deformation (plastic strain is 0). B and C indicate brittle fracture along perpendicular layers showing low strain fracture surface (~0.055). D: ductile fracture in junction layers with sufficient plastic strain (~0.110). E and F show ductile fracture along SiC/Al – Al macro-interface.
Comment #7: I understand that these suggestions may require additional work, but I believe they will contribute to your article's overall quality and impact. I appreciate your effort in this research and look forward to seeing how these suggestions may be integrated into your work.
Response: We thank the Reviewer for the constructive feedback and suggestions. These valuable comments led us to improve the quality of this manuscript. It is obvious that the readability was enhanced.
Comment #8: Comments on the Quality of English Language: The language looks fine.
Response: We thank the Reviewer for the comment on the English. We are excited that the language is approved.

Reviewer 2 Report
Comments and Suggestions for Authors
The manuscript discussed numerically the effect of strength of Sic particles on toughening of metal-matrix composites. For this purpose, particles of different fracture strength in the range of 1-5 GPa have been investigated. In the opinion of this reviewer, the manuscript is recommended for publication considering the following comments.
- It is recommended to use the reference style of the journal in the text.
- The size of the RVE is much higher than the particle length which satisfy the length separation. It is also recommended yet to discuss this in the manuscript.
- From the applied loads, it is clearly that periodic boundary conditions were not satisfied. Could the author discuss why?
- It is recommended to express Eq. (1) in terms of stiffness tensor (K) instead of stress tensor.
- In page 8 lines 255-256, please check if Ref.[11] is a previous work of the authors themselves!.
- In page 4 lines 143-145; it was shown that crack deflection has helped the toughening. What other mechanisms could be part of toughening. Please discuss.
Comments on the Quality of English LanguageMinor editing of English language required
Author Response
Reviewer #2's comments:
Comment #1: The manuscript discussed numerically the effect of strength of SiC particles on toughening of metal-matrix composites. For this purpose, particles of different fracture strength in the range of 1-5 GPa have been investigated. In the opinion of this reviewer, the manuscript is recommended for publication considering the following comments.
Response: We thank the Reviewer for the positive comment. We have revised the manuscript according to the comments and suggestions. The revisions were highlighted RED in the revised manuscript.
Comment #2: It is recommended to use the reference style of the journal in the text.
Response: We thank the Reviewer for the comment on references. The journal name and volume should be italic. We revised the reference style in the Reference List. The revisions were highlighted RED.
Comment #3: The size of the RVE is much higher than the particle length which satisfy the length separation. It is also recommended yet to discuss this in the manuscript.
Response: We thank the Reviewer for the comment on the modelling. In this work, we focused on the composites with network architecture. For this topic, there are a constraint. To enhance the representativeness of geometry model, more network cells and periodic geometry are required. However, it extremely increases the complexity of FE models, which is hard to solve by FEM code. So, a simple network geometry is needed to mimic the real composite structure. Therefore, we built the present 8-cell network composite geometry model without periodicity. We added the consideration on geometry modelling in the Manuscript at Line 85-88 Page 2: “Too many network cells and periodic geometry extremely increase complexity of the models. Thus it requires more and more element mesh, which is hard to be solved by FEM code. So an 8-cell network architecture without periodicity was established”.
Comment #4: From the applied loads, it is clearly that periodic boundary conditions were not satisfied. Could the author discuss why?
Response: We thank the Reviewer for the comment on boundary condition. Honestly, for simulation on representative volume element (RVE) model of homogeneous composites, the periodic boundary condition is the first choice. However, considering on decreasing the complexity of the model, a 8-cell network composite geometry model without periodicity was established. Therefore, the model is not fit to periodic boundary condition. So we kept the boundaries parallel to XOY, YOZ and ZOX planes.
The description on boundary condition was also added in the Manuscript at Line 93-94 Page 2: “Due to lack periodicity in the geometry, the uniaxial tensile boundary condition was applied. These boundary conditions were defined as…”.
Comment #5: It is recommended to express Eq. (1) in terms of stiffness tensor (K) instead of stress tensor.
Response: We thank the Reviewer for the comment on stiffness degradation. The progressive damage degradation behavior in our simulation is presented in the Fig. R4. Obviously, the Young`s modulus (E) degrades after damage:
|
|
(R1) |
where D is the stiffness degradation variable, is the undamaged stiffness tensor. So the stress degradation can be expressed by:
|
|
(R2) |
where σ is stress tensor with degradation and is the undamaged stress tensor.
Fig. R4 Stress-strain curve with progressive damage degradation.
Therefore, the description on Eq. (1): “Thus a progressive degradation of the material stiffness model was applied in the simulation on failure behaviors:
|
|
(1) |
where D is the stiffness degradation variable, σ is stress tensor with degradation and is the undamaged stress tensor” was modified to “Thus a progressive degradation of the material stiffness model was applied in the simulation on failure behaviors[34,35] (Figure S2):
|
|
(1) |
where D is the stiffness degradation variable, E is the Young`s modulus with degradation to represent the stiffness, is the undamaged stiffness tensor. So the stress tensor shows the same form:
|
|
(2) |
where σ is stress tensor with degradation and is the undamaged stress tensor” in the Manuscript at Line 110-115 Page 3.
The Fig. R4 was added in the Supplementary Materials as Fig. S2 with a brief description at Page 2: “The progressive damage degradation behavior is presented in the Fig. S1, which is given by:
|
|
(1) |
where D is the stiffness degradation variable, E is the Young`s modulus with degradation to represent the stiffness, is the undamaged stiffness tensor. So the stress degradation can be expressed by:
|
|
(2) |
where σ is stress tensor with degradation and is the undamaged stress tensor.”
Comment #6: In page 8 lines 255-256, please check if Ref.[11] is a previous work of the authors themselves!
Response: We thank the Reviewer for the comment on the reference citing. We made a mistake on citing in page 8 lines 255-256. It was revised to cite Ref.[16] in the Manuscript.
Comment #7: In page 4 lines 143-145; it was shown that crack deflection has helped the toughening. What other mechanisms could be part of toughening. Please discuss.
Response: We thank the Reviewer for the comment on the toughening mechanisms. The toughening effect is directly related to crack initiation and propagation behaviors.
The micro-crack initiates in the layers parallel to extern load direction, owing to stress concentration. During this process, the mode-I crack dominates the failure behavior of the network composite. The stress at the micro-crack tip (σtip) is related to the crack size (a) and distance to it (r):
|
|
(R3) |
Obviously, the network architecture design means a thin reinforcement-rich layer (low a) and large monolithic matrix (high r). This implies the micro-crack propagation is prevented. So the primary micro-crack tip is blunted by the neighboring matrix cells. Thus composite continues plastic deformation and thus presents a good ductility[R5,R6].
The branching (rectangle in Figure 2b) and deflection (F in Figure 2c) behaviors of macro-crack lead to a zigzag crack surface, and thus longer propagation distance. In addition, zigzag propagation means the crack is not perpendicular to the extern load. Our previous theoretical analysis showed that the crack propagation resistance in-creases dramatically with decreasing orientation between crack and extern load[R7]. These issues cost more energy[R8,R9].
In summary, the network architecture design promotes primary micro-crack blunting, macro-crack deflection and branching. Therefore, the crack propagation is delayed with more energy consumption. This is beneficial to the toughness of the composites.
Therefore, we added the related description in the Manuscript at Line 163-181 Page 4-5: “From viewport of micro-crack, to toughen the architected composite, the large monolithic reinforcement-lean area and thin reinforcement-rich layer are required. During tensile, the mode-I crack dominates the failure behavior of the network composite. The stress at crack tip (σtip) is directly proportional to crack length (a) and inversely proportional to the distance to it (r):
|
|
(6) |
Due to stress concentration in the reinforcement-rich layer, the micro-crack initiates in the layers parallel to extern load direction. The length of mode-I micro-crack depends on the thickness of the layer. A large monolithic reinforcement-lean area implies increasing r. Thus, the primary micro-crack in network layer tends to be blunted by neighboring matrix cells (circle in Figure 2b). From the viewport of macro-crack, the crack branching (rectangle in Figure 2b) and deflection (F in Figure 2c) lead to a zigzag crack morphology, and thus longer propagation distance. These behaviors cost more energy. In addition, zigzag propagation means the crack is not perpendicular to the extern load. Our previous theoretical analysis showed that the crack propagation resistance increases dramatically with decreasing orientation between crack and extern load[38]. This analysis is fixed with the experimental observations that the network architecture design promotes primary micro-crack blunting[42,43], macro-crack deflection[44,45] and branching[44,45]. Therefore, the crack propagation is delayed with more energy consumption. This is beneficial to the toughness of the composites.”
The references were added in the Reference List:
“42. Zhang, W.; Wang, M.; Chen, W.; Feng, Y.; Yu, Y. Preparation of TiBw/Tie6Ale4V composite with an inhomogeneous re-inforced structure by a canned hot extrusion process. J. Alloys Compd. 2016, 669, 79–90. http://doi.org/10.1016/j.jallcom.2016.01.228
- Liu, G.F.; Chen, T. J. Enhancement of strength–ductility synergy in heterostructured 2024Al alloy through tailoring het-erogeneity level. J. Mater. Sci. 2023, 58, 11820–11839. https://doi.org/10.1007/s10853-023-08752-5
- Rong, X.; Zhang, X.; Zhao, D.; He, C.; Shi, C.; Liu, E.; Zhao, N. In-situ Al2O3 -Al interface contribution towards the strength-ductility synergy of Al-CuO composite fabricated by solid-state reactive sintering. Scr. Mater. 2021, 198, 113825. https://doi.org/10.1016/j.scriptamat.2021.113825
- Zhang, F.; Wang, J.; Liu, T.; Shang, C. Enhanced mechanical properties of few-layer graphene reinforced titanium alloy matrix nanocomposites with a network architecture. Mater. Des. 2020, 186, 108330. https://doi.org/10.1016/j.matdes.2019.108330”
Comment #8: Comments on the Quality of English Language:Minor editing of English language required.
Response: We thank the Reviewer for the comment on the English. We polished the English with minor editing.
At Line 40-41 Page 1: “Utilizing X-ray CT data and finite element method (FEM), it was indicated that…” was modified to “The X-ray CT data and finite element method (FEM) indicated that…”
At Line 57-58 Page 2: “However, applying SiC particles various of fracture strength in the experience require much time consumption.” was modified to “However, applying SiC particles with various of fracture strength in the experience requires much more time consumption.”
At Line 197-198 Page 5: “...further increasing SiC strength cannot further increase the composite strength because...” was modified to “...further increasing SiC strength cannot enhance the composite strength because...”
At Line 199-200 Page 5: “Compared the stress-strain curves of composites with network and homogeneous particle distribution...” was modified to “Comparing the stress-strain curves of composites with network and homogeneous distributions of particle...”
At Line 205-207 Page 6: “So, more load was transferred from matrix to particles. And the load-bearing efficiency of network-arranged reinforcement (see Figure S3) is high.” was modified to “As a result, more load was transferred from matrix to particles, and the load-bearing efficiency of network-arranged reinforcement (see Figure S3) is improved.”
At Line 209-210 Page 6: “Moreover, the increased UTS in network composite often sacrifices the elongation and thus toughness...” was modified to “Moreover, the increased UTS in network composite often sacrifices the elongation and toughness...”
At Line 350-351 Page 10: “The threshold particle strength existed and was different for homogeneous and network composites.” was modified to “The threshold strength of particle was different between homogeneous and network composites.”

Round 2
Reviewer 1 Report
Comments and Suggestions for Authors
Suggestions and corrections are incorporated.